# Statistical estimation of global surface temperature response to forcing under the assumption of temporal scaling

Eirik Myrvoll-Nilsen[1], Sigrunn Holbek Sørbye[1], Hege-Beate Fredriksen[1], Håvard Rue[2], and Martin Rypdal[1]

[1]Department of Mathematics and Statistics, UiT The Arctic University of Norway, N-9037 Tromsø, Norway
[2]CEMSE Division, King Abdullah University of Science and Technology, Thuwal, Saudi Arabia

**Correspondence:** Eirik Myrvoll-Nilsen (eirik.myrvoll-nilsen@uit.no)

**Abstract.** Reliable quantification of the global mean surface temperature (GMST) response to radiative forcing is essential for assessing the risk of dangerous anthropogenic climate change. We present the statistical foundations for an observation-based approach, using a stochastic linear-response model that is consistent with the long-range temporal dependence observed in global temperature variability. We have incorporated the model in a latent Gaussian modeling framework, which allows for the use of integrated nested Laplace approximations (INLAs) to perform full Bayesian analysis. As examples of applications, we estimate the GMST response to forcing from historical data and compute temperature trajectories under the Representative Concentration Pathways (RCPs) for future greenhouse gas forcing. For historic runs in the Model Intercomparison Project Phase 5 (CMIP5) ensemble, we estimate response functions and demonstrate that one can infer the transient climate response (TCR) from the instrumental temperature record. We illustrate the effect of long-range dependence by comparing the results with those obtained from one-box and two-box energy balance models. The software developed to perform the given analyses is publicly available as the `R-package INLA.climate`.

## 1 Introduction

Despite decades of research and development of global circulation models (GCMs) and Earth system models (ESMs), the discrepancies between models remain substantial, even as we describe physical processes with increasing accuracy and resolution. Part of the model spread is associated with a lack of understanding of the shortwave cloud feedback (Qu et al., 2018). However, there are several other modeling choices and compromises that contribute to the uncertainty (Flato, 2011). As a consequence, several studies have focused on constraining model results on climate sensitivity on observational data, see e.g., the work of Annan and Hargreaves (2006), or the more recent studies of Cox et al. (2018) and Rypdal et al. (2018b, a). These studies focus on the equilibrium climate sensitivity (ECS) as an essential metric of the climate response, as have numerous paleoclimate studies (Hansen et al., 2013; von der Heydt and Ashwin, 2017; Köhler et al., 2017).

A simpler approach is to adopt a linear approximation, and to apply statistical methods to extract information on the climate response from data on global surface temperature and radiative forcing in the instrumental era. Under the assumption of a linear and stationary response, the global surface temperature anomaly $\Delta T$ can be expressed as a filtering of the global radiative

forcing $F$, mathematically expressed as

$$\Delta T(t) = \int_{-\infty}^{t} G(t-s)\Big( F(s)ds + \sigma dB(s) \Big), \tag{1}$$

where $\sigma dB(t)$ represents a white-noise forcing that gives rise to internal climate variability, and $G$ is the response function, or Green's function, characterizing the relation between forcing and the temperature anomaly. A model of this form can arise from the simplest energy-balance model, i.e. the equations

$$\frac{d\Delta Q}{dt} = -\lambda \Delta T + F \tag{2}$$

and $\Delta Q = C\Delta T$, where $\Delta Q$ is the change in the system's heat content corresponding to a temperature change $\Delta T$, and $C$ is a heat capacity (Rypdal, 2012). If a white-noise forcing term is included on the right-hand side of Eq. (2) it becomes a stochastic differential equation with a stationary solution on the form of Eq. (1), with $G(t) = C^{-1}e^{-t/\tau}$ and $\tau = C/\lambda$. The process has a natural decomposition into the response to the known forcing,

$$\Delta T_{\text{det}}(t) = \frac{1}{C} \int_{-\infty}^{t} e^{-(t-s)/\tau} F(s)ds \tag{3}$$

and a stochastic term

$$X(t) = \frac{\sigma}{C} \int_{-\infty}^{t} e^{-(t-s)/\tau} dB(s), \tag{4}$$

which for this particular model is an Ornstein-Uhlenbeck process. Rypdal and Rypdal (2014) show how the parameters of the two terms can be estimated simultaneously from time series of forcing and the GMST using the maximum likelihood (ML) method. They also demonstrate that the resulting process is inconsistent with observations. The stochastic term $X(t)$ does not exhibit the strong positive decadal-scale serial correlations that is observed in the GMST in the instrumental era, and secondly, the model's response to reconstructed forcing for the last millennium does not show sufficient low-frequency variability compared to Northern-hemisphere temperature reconstructions.

The inconsistency of the simple energy-balance model is due to the slow climate response associated with the energy exchange with the deep ocean. One can easily incorporate this effect within the framework of Eq. (1), by generalizing the zero-dimensional (one-box) model to a two-box model that includes a layer representing the deep ocean (Geoffroy et al., 2013; Held et al., 2010; Caldeira and Myhrvold, 2013), or the more general $m$-box model discussed by Fredriksen and Rypdal (2017).

The generalization from the zero-dimensional (one-box) energy balance model to the two-box model, or the $m$-box models, means that the number of free parameters increases. Concerning statistical inference, this could be problematic due to potential over-fitting. Mathematically, the generalization of Eq. (2) is on the form

$$\mathbf{C}\frac{d\Delta \mathbf{T}(t)}{dt} = \mathbf{K}\Delta \mathbf{T}(t) + \mathbf{F}(t), \tag{5}$$

where the diagonal elements of $\mathbf{C}$ are the heat capacities of each box, and the matrix $\mathbf{K}$ contains coefficients describing heat exchange between boxes and the feedback parameter $\lambda$.

The system in Eq. (5) is solved by bringing the matrix $\mathbf{C}^{-1}\mathbf{K}$ to diagonal form, and the surface temperature anomaly can be written as in Eq. (1) where $G(t)$ is the weighted sum of $m$ exponential functions (Fredriksen and Rypdal, 2017):

$$G(t) = \begin{cases} \sum_{k=1}^{m} w_k e^{-t/\tau_k}, & t \geq 0 \\ 0, & t < 0 \end{cases}, \tag{6}$$

The characteristic time scales $\tau_k = -1/\mu_k$ are defined from the eigenvalues $\mu_k$ of $\mathbf{C}^{-1}\mathbf{K}$ and $w_k$ denotes the weight of the $k$th exponential function.

On the other hand, global temperature variability exhibits a scaling symmetry. For instance, both the forced and the unforced global temperature variability have power spectral densities (PSDs) that are approximate power laws,

$$S(f) \sim f^{-\beta} \tag{7}$$

for frequencies corresponding to time scales ranging from months to centuries (Rypdal and Rypdal, 2016; Rybski et al., 2006; Lovejoy and Schertzer, 2013; Huybers and Curry, 2005; Franzke, 2010; Fredriksen and Rypdal, 2016). The global temperature fluctuations are consistent with a fractional Gaussian noise (fGn), which can formally be defined by the integral analogous to Eq. (4), but with the exponential response function replaced with a scale-invariant response function

$$G(t) = \left(\frac{t}{\gamma}\right)^{\beta/2-1} \xi. \tag{8}$$

Here, $\gamma$ is a scale parameter with the dimension of time, and $\xi$ is a variable needed in order for $G(t)$ to have the correct physical dimensions. The scaling exponent $\beta$ (defined from the PSD in Eq. (7)) relates to the so-called Hurst exponent of the fGn via the formula $\beta = 2H - 1$. Based on this Rypdal and Rypdal (2014) proposed a fractional linear response model in the form of Eq. (1), in which the parsimonious expression in Eq. (8) replaces the linear combination of exponential functions in Eq. (6). The cost of the reduction in model complexity is that the fractional linear response model does not have a well-defined ECS, and in general, we cannot write the model as a system of differential equations as in Eq. (5). But on time scales up to approximately $10^3$ years, the model provides an accurate description of both forced and unforced surface temperature fluctuations (Rypdal and Rypdal, 2014; Rypdal et al., 2015), and the millennial-scale climate sensitivity in the estimated fractional linear response model correlates strongly with ECS over the ensemble of models in the Coupled Model Intercomparison Project Phase 5 (CMIP5) (Rypdal et al., 2018a).

Temporal scale invariance in global temperature fluctuations is an empirical observation and we cannot deduce the parameters in the fractional linear response model from physical principles. This paper presents a statistical methodology that makes it possible to fit the model to observational data and estimate all model parameters. Parameter estimation is done within a Bayesian framework making use of the methodology of integrated nested Laplace approximation (INLA) for latent Gaussian models introduced in Rue et al. (2009). Barboza et al. (2019) use this framework to investigate model formulations and forcing components in paleoclimate reconstructions.

The INLA-methodology and inference for our statistical model, assuming the scale-invariant response function in Eq. (8), is described further in Section 2. This section explains how to compute the marginal posterior distributions of the model parameters. As the model has a non-standard form, this includes certain modifications of the INLA-methodology to ensure computational efficiency. We discuss applications in Section 3.

In Section 3.1 we fit the model to the temperature and forcing data set generated by the GISS-E2-R (Schmidt et al., 2014) ESM. Here we show how to extract the GMST response to the known forcing using a Monte Carlo sampling approach. In Section 3.2, the model is used for temperature forecasting where the representative concentration pathway (RCP) trajectories describe the future CO2 forcing. Section 3.3 describes how the transient climate response (TCR) can be estimated using our model. We obtain estimates for 19 temperature series and their associated adjusted forcing series.

We compare the resulting estimates of TCRs with the TCRs obtained directly from the respective ESMs, and with TCR estimates from historical HadCRUT4 temperature data set using different forcing data. The applications are incorporated in the R-package INLA.climate. This package also includes the option of using the exponential response functions as defined by Eq. (3) and Eq. (4) for the one-box model and Eq. (6) for $m$-box models. A discussion and final conclusions are given in Section 4.

## 2 Discrete-time modeling and statistical inference

Rypdal and Rypdal (2014) use an ML estimator to estimate the model parameters from the observational yearly time series of $\mathbf{\Delta T} = (\Delta T_1, \dots, \Delta T_n)$ of GMST, and the corresponding vector of radiative forcing $\mathbf{F} = (F_1, \dots, F_n)$. Here, we estimate parameters by adopting a Bayesian framework, making use of the INLA-methodology (Rue et al., 2009, 2017). This approach implies that parameters are treated as stochastic variables and assigned prior distributions. The information given by the priors is then combined with the likelihood of the observations and updated to give posterior distributions using Bayes' theorem.

In a discrete-time model, we assume that $\Delta T_t$ has a Gaussian distribution with a random mean expressed by the linear predictor

$$\eta_t = \sigma_f \sum_{s=1}^{t} G_{ts}(H)\big(F_s + F_0\big) + \varepsilon_t, \quad t = 1, \dots, n \tag{9}$$

where $\sigma_f = \gamma^{-\beta/2+1}$ while $F_0$ denotes a shift parameter which gives the initial forcing value. $G_{ts}$ denotes a discretely indexed element of the function,

$$G_{ts}(H) = \begin{cases} (t - s + \frac{1}{2})^{H-\frac{3}{2}}, & 1 \leq s \leq t \leq n \\ 0, & \text{otherwise} \end{cases}. \tag{10}$$

Further, the vector $\boldsymbol{\varepsilon} = (\varepsilon_1, \dots, \varepsilon_n)$ denotes a zero-mean fGn process, implying that the covariance between $\varepsilon_t$ and $\varepsilon_s$ is

$$\Sigma_{ts} = \frac{\sigma_\varepsilon^2}{2}\big(|t - s + 1|^{2H} + |t - s - 1|^{2H}$$
$$- 2|t - s|^{2H}\big), \quad t, s = 1, \dots, n, \tag{11}$$

where $\sigma_\varepsilon = \sigma \sigma_f$. In vector notation, the predictor is then given by

$$\boldsymbol{\eta} = \boldsymbol{\mu} + \boldsymbol{\varepsilon} \tag{12}$$

where

$$\boldsymbol{\mu} = \boldsymbol{\mu}(H, \sigma_f, F_0) = \sigma_f \mathbf{G}(H)(\mathbf{F} + F_0). \tag{13}$$

The covariance matrix of the predictor is $\boldsymbol{\Sigma} = \boldsymbol{\Sigma}(H, \sigma_\epsilon)$ with the elements in Eq. (11). Notice that the matrix $\mathbf{G}(H)$ is lower

triangular with elements given by Eq. (10). The given formulation implies that the vector $\boldsymbol{\mu}$ represents the GMST response to the known forcing $\mathbf{F}$ while $\boldsymbol{\varepsilon}$ is the GMST response to the random forcing, i.e. the unforced climate variability.

The statistical regression formulation in Eq. (9) has a hierarchical structure in which the expected temperature anomalies are modelled in terms of the random predictor $\boldsymbol{\eta}$ with elements specified by Eq. (9). The predictor depends on additional model parameters $\boldsymbol{\theta} = (H, \sigma_\varepsilon, \sigma_f, F_0)$. This set-up implies that we need to assign priors, both to the predictor and to the

model parameters. By assigning a Gaussian prior to $\boldsymbol{\eta}$, the resulting model becomes a latent Gaussian model, which can be analyzed using the INLA-methodology. In general, this class of models introduces a latent Gaussian field $\boldsymbol{x}$, which contains all the random components of a linear predictor, including the predictor itself. In our case, the latent field is equal to the linear predictor, $\boldsymbol{x} = \boldsymbol{\eta} = \boldsymbol{\mu} + \boldsymbol{\varepsilon}$. However, inference for this model is not straightforward as the model parameter $H$ appears in both of the terms $\boldsymbol{\mu}$ and $\boldsymbol{\varepsilon}$. We choose to circumvent this problem by considering the sum $\boldsymbol{\mu} + \boldsymbol{\varepsilon}$ as a single model component, i.e., as

a fractional Gaussian noise process with mean vector $\boldsymbol{\mu}$ and covariance matrix $\boldsymbol{\Sigma}$. The dependence between two components implies that we will not get separate posterior estimates for $\boldsymbol{\mu}$ and $\boldsymbol{\varepsilon}$, directly.

Using $p(\cdot)$ as a generic notation for probability density functions, we can summarize the three-stage hierarchical structure of latent Gaussian models, including distributional assumptions, as follows:

– The first stage specifies the likelihood of the model. The observed temperature anomaly $\Delta T_t$ is assigned a Gaussian
distribution with negligible fixed variance and mean $\eta_t$. The observations are assumed to be conditionally independent given the latent field $\mathbf{x}$ and parameters $\boldsymbol{\theta}$, i.e.

$$p(\Delta \mathbf{T} \mid \mathbf{x}, \boldsymbol{\theta}) = \prod_{t=1}^{n} p(\Delta T_t \mid x_t, \boldsymbol{\theta}).$$

– The second stage specifies the prior distribution for the latent field. Given the parameters $\boldsymbol{\theta}$, the latent field $\mathbf{x}$ is assigned a Gaussian prior distribution with mean vector $\boldsymbol{\mu}_x = \mathrm{E}[\mathbf{x} \mid \boldsymbol{\theta}]$ and precision matrix, $\mathbf{Q} = \mathbf{Q}(H, \sigma_\varepsilon)$, defined as the inverse
covariance matrix, i.e.

$$p(\mathbf{x} \mid \boldsymbol{\theta}) = \sqrt{\frac{\det \mathbf{Q}}{(2\pi)^n}} \exp\left(-\frac{1}{2}(\mathbf{x} - \boldsymbol{\mu}_x)^T \mathbf{Q}(\mathbf{x} - \boldsymbol{\mu_x})\right).$$

– The third stage specifies independent priors for the parameters:

$$p(\boldsymbol{\theta}) = p(H)p(\sigma_\varepsilon)p(\sigma_f)p(F_0).$$

The shift parameter $F_0$ is assigned a zero-mean Gaussian prior, while the other parameters are assigned penalised complexity (PC) priors (Simpson et al., 2017). The class of PC priors represents a recently developed framework to compute priors based on specific principles, including support to Occam's razor. The PC prior of the two scaling parameters $\sigma_f$ and $\sigma_\epsilon$ can be computed to equal the exponential distribution while the PC prior of $H$ is computed numerically (Sørbye and Rue, 2018).

The joint posterior for all components of the latent field and all of the model parameters is then summarized by

$$p(\mathbf{x}, \boldsymbol{\theta} \mid \Delta \mathbf{T}) \propto \prod_{t=1}^{n} p(\Delta T_t \mid x_t, \boldsymbol{\theta}) p(\mathbf{x} \mid \boldsymbol{\theta}) p(\boldsymbol{\theta}).$$

Our main objective is to estimate the marginal posterior distribution for all components of the latent field

$$p(x_t \mid \Delta \mathbf{T}) = \int p(x_t \mid \boldsymbol{\theta}, \Delta \mathbf{T}) p(\boldsymbol{\theta} \mid \Delta \mathbf{T}) \mathrm{d}\boldsymbol{\theta}, \quad t = 1, \ldots, n \tag{14}$$

and the marginal posteriors for all the model parameters

$$p(\theta_j \mid \Delta \mathbf{T}) = \int p(\boldsymbol{\theta} \mid \Delta \mathbf{T}) \mathrm{d}\boldsymbol{\theta}_{-j}, \quad j = 1, \ldots, 4. \tag{15}$$

Here, the notation $\boldsymbol{\theta}_{-j}$ is used to denote the vector $\boldsymbol{\theta}$ excluding the $j$th parameter. The posterior distributions provide a complete description of the latent field components and the parameters in our model. From the marginals in Eq. (14)–Eq. (15) we can extract summary statistics such as the mean, variance, quantiles and credible intervals.

Traditionally, marginal posterior distributions have been approximated using Markov chain Monte Carlo methods (Robert and Casella, 1999). Such methods are simulation-based and can potentially be very time-consuming for hierarchical models. The INLA-methodology represents a computationally superior, but still accurate, alternative and is available using the R-package R-INLA. This package can be downloaded for free at www.r-inla.org. INLA provides a deterministic approach, approximating the posterior distributions in Eq. (14)–Eq. (15) using numerical optimization techniques, interpolations and numerical integration. Among others, this includes the use of the Laplace approximation (Tierney and Kadane, 1986) which is an old technique to compute high-dimensional integrals. Specifically, the joint posterior distribution for the model parameters in Eq. (15) is approximated by employing a Laplace approximation evaluated at the mode $\mathbf{x}^*(\boldsymbol{\theta})$:

$$p(\boldsymbol{\theta} \mid \Delta \mathbf{T}) \approx \left. \frac{p(\mathbf{x}, \boldsymbol{\theta}, \Delta \mathbf{T})}{p_G(\mathbf{x} \mid \boldsymbol{\theta}, \Delta \mathbf{T})} \right|_{\mathbf{x} = \mathbf{x}^*(\theta)}, \tag{16}$$

where $p_G(\mathbf{x} \mid \boldsymbol{\theta}, \Delta \mathbf{T})$ is a Gaussian approximation of

$$p(\mathbf{x} \mid \boldsymbol{\theta}, \Delta \mathbf{T}) \propto p(\mathbf{x} \mid \boldsymbol{\theta}) p(\Delta \mathbf{T} \mid \mathbf{x}, \boldsymbol{\theta}).$$

This approximation is usually very accurate as we know that $p(\mathbf{x} \mid \boldsymbol{\theta})$ is already Gaussian. The marginal for each model parameter is then obtained by assuming a normal distribution modified to allow for skewness,

$$p(\theta_j \mid \Delta \mathbf{T}) \approx \begin{cases} \mathcal{N}(0, \sigma_{j+}^2), & \theta_j > 0 \\ \mathcal{N}(0, \sigma_{j-}^2), & \theta_j \leq 0 \end{cases}.$$

The scaling parameters $\sigma_{j+}$ and $\sigma_{j-}$ are found using the approximate joint posterior distribution of Eq. (16), see Martins et al. (2013) for details. To compute Eq. (14), the Laplace approximation in Eq. (16) is combined with a simplified and computationally faster version of the Laplace approximation of $p(x_t \mid \boldsymbol{\theta}, \Delta\mathbf{T})$. Finally, the integrand of Eq. (14) is evaluated for values of $\boldsymbol{\theta}$ in a grid efficiently covering the parameter space for $\boldsymbol{\theta}$, see Rue et al. (2009) and Rue et al. (2017) for details.

A key assumption for the numerical approximations to be computationally efficient is that the latent Gaussian field $\mathbf{x}$ has Markov properties, i.e. $\mathbf{x}$ needs to be a Gaussian Markov random field having a sparse precision matrix $\boldsymbol{Q}$ (Rue and Held, 2005). This is not the case for fGn as the long-range dependency structure of this process gives a dense precision matrix. We resolve this problem by approximating $\boldsymbol{\varepsilon}$ as a weighted sum of $m$ independent first-order autoregressive (AR(1)) processes, i.e.

$$\tilde{\boldsymbol{\varepsilon}} = \sum_{i=1}^{m} \sqrt{w_i}\,\tilde{\mathbf{x}}_i. \tag{17}$$

To capture the correlation structure between $\tilde{\boldsymbol{\varepsilon}}$ and each of the AR(1) processes $\tilde{\mathbf{x}}_i$, the latent field must be extended to also include the underlying AR(1) processes, i.e. $\mathbf{x} = (\boldsymbol{\eta}, \boldsymbol{\mu} + \tilde{\boldsymbol{\varepsilon}}, \tilde{\mathbf{x}}_1, \ldots, \tilde{\mathbf{x}}_m)$. The weights $\{w_i\}_{i=1}^{m}$ and the first-lag autocorrelation coefficients of the AR(1) processes are selected such that the resulting autocorrelation function of $\tilde{\boldsymbol{\varepsilon}}$ best approximates that of fGn. In addition to ensuring computational efficiency, this approximation also proves to be remarkably accurate. For further details about this approximation, see Sørbye et al. (2019) who also provide a discussion from a statistical perspective. For a physical interpretation of this approximation we refer to Fredriksen and Rypdal (2017).

Currently, there are no built-in model components in `R-INLA` which suit our specifications. This means that we have to construct one manually using `rgeneric`, a modeling tool that allows generic model components to be defined for INLA. To make this accessible to applied scientists we have developed an R-package called `INLA.climate` which includes functions that take care of the technical part of the fitting procedure and presents important information and summary statistics in a readable format. This package contains a versatile and user-friendly interface to fit the model in Eq. (9) and includes functions to replicate all results presented in this paper. The package is available at the GitHub repository `www.github.com/eirikmn/INLA.climate`. Detailed description of the package and its features is available in its accompanying documentation.

## 3  Applications

### 3.1  Estimating the forced temperature response

As explained in Section 2, our model formulation implies that the sum $\boldsymbol{\mu} + \boldsymbol{\varepsilon}$ is viewed as one model component. Consequently, INLA will give an estimate for the posterior distribution of the sum, and not the marginal posterior distributions for each of the terms $\boldsymbol{\mu}$ and $\boldsymbol{\varepsilon}$.

In this example, we illustrate how we can approximate the marginal posterior distribution for the temperature response attributed to the known forcing, $p(\mu_i \mid \Delta\mathbf{T})$, combining INLA with Monte Carlo sampling. We first fit our model to the GISS-E2-R temperature and the corresponding forcing data, using INLA. This gives the estimated marginal posterior distributions for each of the model parameters $\boldsymbol{\theta} = (H, \sigma_\varepsilon, \sigma_f, F_0)$, as shown in Fig. 1. Next, we use the `inla.hyperpar.sample`

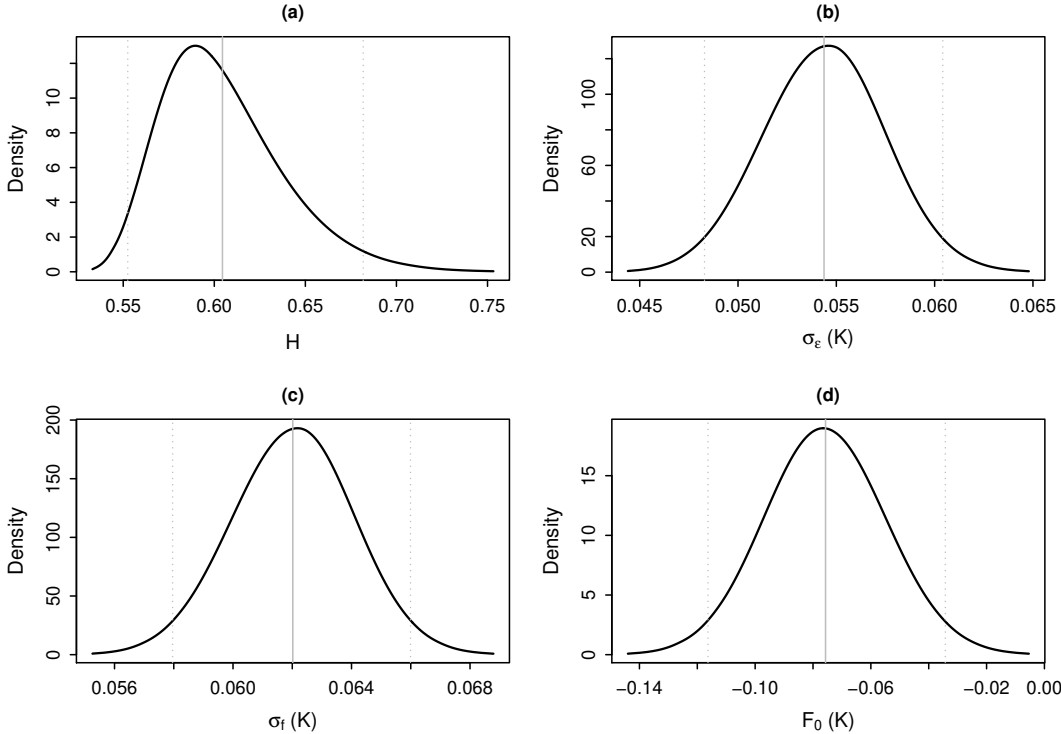

**Figure 1.** The marginal posterior distributions of the parameters, obtained using `INLA.climate` to fit our model to the GISS-E2-R temperature and forcing dataset. The vertical lines show the mean and 95 % credible intervals.

function from the `R-INLA` package to draw 100,000 samples from the approximate joint posterior distribution of $H, \sigma_f$ and
$F_0$. For each of these samples, we compute $\boldsymbol{\mu}$ according to Eq. (13). The resulting samples give approximate marginal posterior distributions for each $\mu_i$ which can then be used to calculate summary statistics.

For comparison, we apply the same approach to estimate the given marginal posterior distributions under the assumption of an exponential response function in Eq. (3). In this case, the discretized unforced response described in Eq. (4) is an AR(1) process. More generally, the discretized response functions corresponding to Eq. (6) are a weighted sum of $m$ AR(1) processes.
Notice that a mixture of a few AR(1) processes will in general have short-range dependence properties, while the approximation in Eq. (17) is constructed to exihibit the long-range dependency structure of fGn. Using the scale-invariant response function or the exponential response functions corresponding to the one- and two-box models, we can compute the marginal posterior means and 95 % credible intervals for each $\mu_i$. The results are shown in Fig. 2. The marginal posterior means are very similar. However, we observe significantly wider credible intervals for the model using the single exponential response function. The
larger uncertainty suggests that a smaller portion of the variance is explained by the unforced climate variability, leaving more of the variation to be explained by the response to the known forcing. Using the `INLA.climate` package, we obtain full inference in seconds on a personal computer. The code to run the example is as follows:

```
        data("GISS_E2_R")
        y = GISS_E2_R$Temperature
215     z = GISS_E2_R$Forcing
        r.scaling = inla.climate(y,z,compute.mu="full")
        r.exponential = inla.climate(y,z,compute.mu="full",m=1,model="ar1")
        r.2exponential = inla.climate(y,z,compute.mu="full",m=2,model="ar1")
```

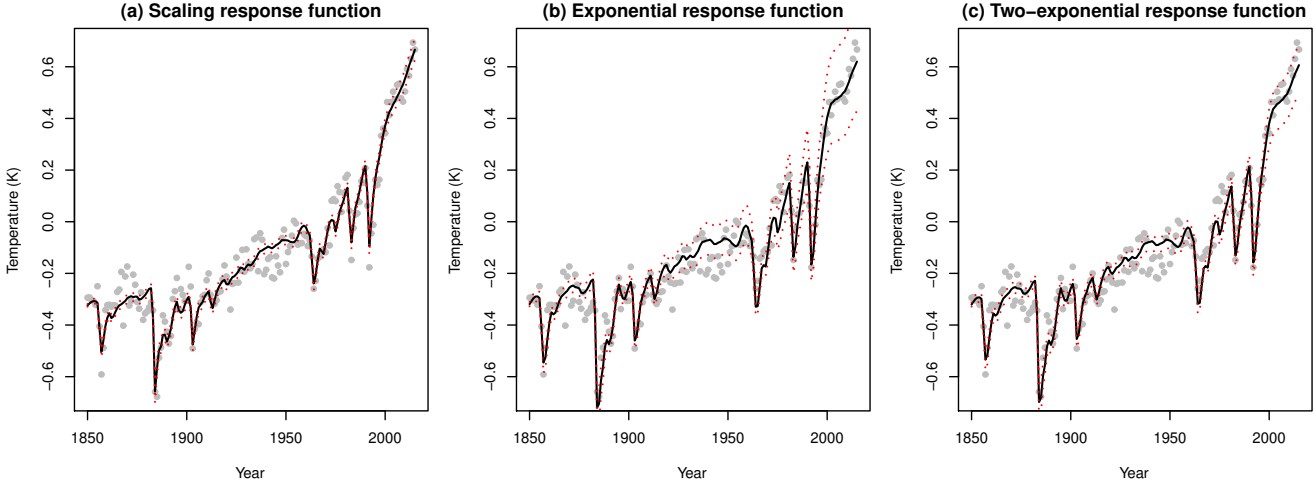

**Figure 2.** The marginal posterior mean and 95 % credible intervals of the temperature response to known forcing $\mu$ obtained by 100,000 Monte Carlo simulations compared to GISS-E2-R temperature data. Panel (a) shows the results under the scale-invariant assumption, panel (b) shows the results using a single exponential response, and panel (c) shows the results using a mixture of two exponential response functions.

### 3.2 Temperature predictions for Representative Concentration Pathway trajectories

Once trained on historical temperature and forcing data, the response model can easily be used to obtain temperature predictions for different future forcing scenarios. Here, we present global temperature predictions for the years 2016 to 2100 based on the HADCRUT4 temperature data and the greenhouse gas component of the Hansen forcing data for 1850 to 2015. For future forcing, we use the representative concentration pathways (RCPs), RCP2.6 (van Vuuren et al., 2007), RCP4.5 (Clarke et al., 2007; Smith and Wigley, 2006; Wise et al., 2009), RCP6 (Fujino et al., 2006; Hijioka et al., 2008) and RCP8.5 Riahi et al.
(2007). These trajectories were first published in 2000 and cover the years 2000 to 2100. In our analyses, we use the RCP for the year 2016 to the year 2100 and adjust each of them so that the forcing in 2015 equals the greenhouse gas forcing in Hansen data in 2015. The forcing scenarios are shown in Fig. 3.

Prediction is carried out using INLA.climate by appending the future scenario to the forcing of the past $\mathbf{F} = (\mathbf{F}_{\text{past}}, \mathbf{F}_{\text{future}})$. The package automatically replaces missing observations with NA values and give predictions for these based on the model
fitted to the observed data.

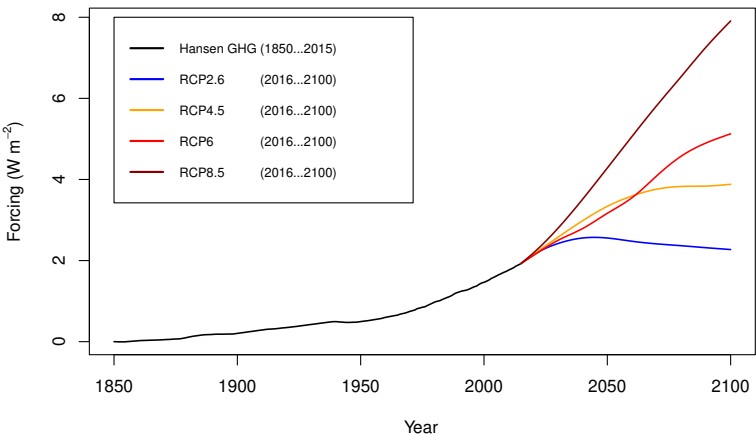

**Figure 3.** The greenhouse gas component of the Hansen forcing (black) followed by the RCP2.6 (blue), RCP4.5 (orange), RCP6 (red) and RCP8.5 (dark red) forcing scenarios.

As in the previous example, we compare the results using the scale-invariant response function versus the exponential response functions corresponding to the one- and two-box models. Training and predictions only take seconds to carry out on a personal computer. Fig. 4 shows the marginal posterior means and 95 % credible intervals using the scale-invariant response, while Fig. 5- 6 show the corresponding results using the exponential response functions. The figures also show
comparisons with the AR5 projections listed in table SPM.2 in IPCC (2013b). We observe that both of the exponential response models seem to fail in describing the persistence in the temperature response and underestimate the global warming increase projections. The predictions obtained using the scale-invariant response function give higher future temperatures estimates, slightly overestimating the AR5 projections.

In Fig. 7 we compare the scale-invariant model's temperature projections for RCP scenarios to ESM temperature projections.
In this analysis, we tune the statistical model to historical runs of different EMSs in the CMIP5 ensemble. As in the other analyses, we use model-specific forcing for each of the ESM. Using the same method as for historical forcing, we also estimate model-specific RCP forcing for each of the ESMs, and each RCP scenario. From this RCP forcing, and using the estimated parameters, we project GMST under the RCP scenarios and compare with the corresponding projections in the ESMs. Fig. 7. shows the differences between the two types of projections. As seen from the figure, there is a negative trend for almost all
models and all scenarios, indicating that the predictions made using the statistical model slightly overestimate the temperature increase in the ESMs. This overestimation is not a statistical bias. Instead, it shows that scale invariance is too crude an approximation for several ESMs. One can interpret the differences as a weakness of the scale-free model. However, not all climate models have scaling properties consistent with temperature reconstructions, and it could also reflect the weaknesses of the ESMs.

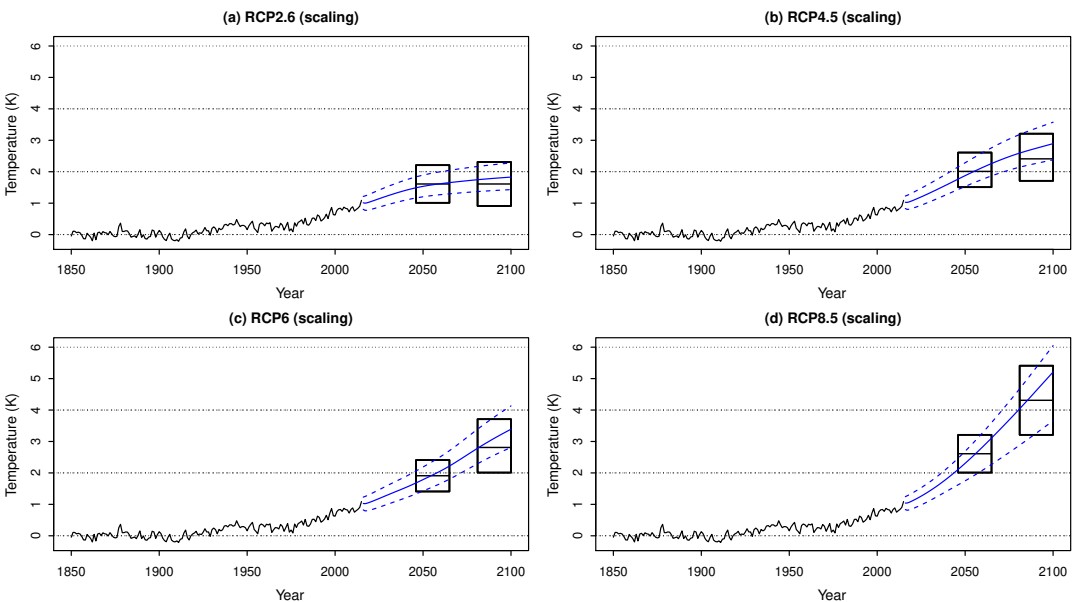

**Figure 4.** Panels (a)–(d) describe the marginal posterior means and 95 % credible intervals of the predicted temperature response to future forcing according to the RCP2.6, RCP4.5, RCP6 and RCP8.5 trajectories, respectively, using a scaling response function. These are compared to the AR5 projections (black boxes).

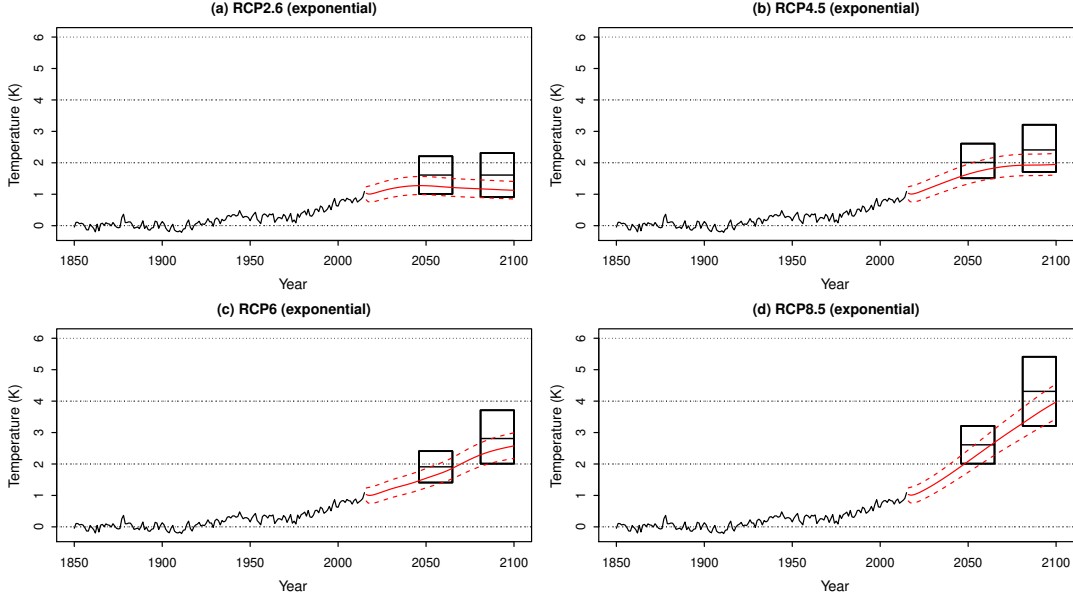

**Figure 5.** Panels (a)–(d) describe the marginal posterior means and 95 % credible intervals of the predicted temperature response to future forcing according to the RCP2.6, RCP4.5, RCP6 and RCP8.5 trajectories, respectively, using a single exponential response function. These are compared to the AR5 projections (black boxes).

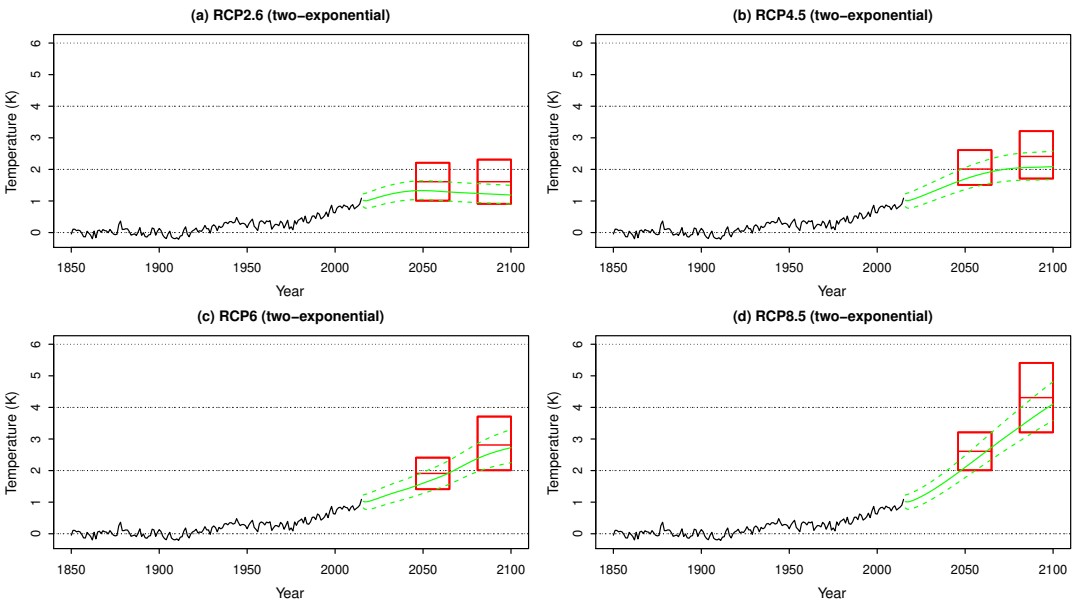

**Figure 6.** Panels (a)–(d) describe the marginal posterior means and 95 % credible intervals of the predicted temperature response to future forcing according to the RCP2.6, RCP4.5, RCP6 and RCP8.5 trajectories, respectively, using a mixture of two exponential response functions. These are compared to the AR5 projections (black boxes).

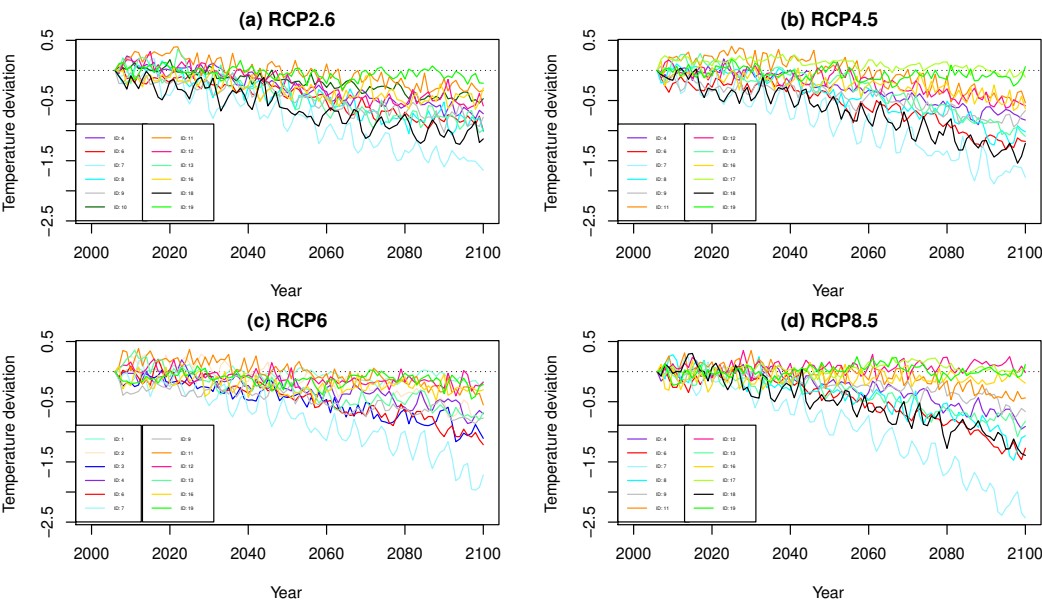

**Figure 7.** Panels (a)–(d) describe the deviations between the predicted marginal posterior means and the predictions obtained from each ESM.

## 3.3 Estimating the transient climate response

As a final application, we describe how our suggested method using a scale-invariant response function can be used to estimate the TCR. The TCR is defined as the average temperature response between 60 and 80 years following a gradual $CO_2$-doubling, assuming a 1 % annual increase. In this scenario, the forcing increases linearly according to

$$f(s) = \frac{Q_{2\times CO_2}}{70 \text{ yrs}}(s + F_0), \quad \text{for } s = 1, ..., 80 \text{ yrs.}$$

Here, $Q_{2\times CO_2}$ is a model-specific coefficient describing the forcing corresponding to a $CO_2$ doubling. We obtain these coefficients from Forster et al. (2013) for all ESMs analyzed in this paper.

The computation of TCR is carried out by inserting the forcing into Eq. (1) and performing the matrix multiplication

$$\mathbf{v} = \sigma_f \mathbf{G}(H)\mathbf{f}.$$

where $\mathbf{G}(H)$ is the $80 \times 80$ matrix with elements defined as in Eq. (10) and $\mathbf{f} = \big(f(1), ..., f(80)\big)^\top$. This implies that TCR is computed by

$$\text{TCR} = \frac{1}{20 \text{ yrs}} \sum_{t=60 \text{ yrs}}^{80 \text{ yrs}} v_t. \tag{18}$$

As in Section 3.1, the approximate marginal posterior distribution for TCR is obtained by combining INLA with Monte Carlo sampling. We first generate samples from the joint posterior distribution of the model parameters $p(\boldsymbol{\theta} \mid \Delta\mathbf{T})$. For each of these samples, we calculate TCR, which then gives the approximate posterior distribution for TCR.

For our analyses, we use temperature data sets generated from 19 ESMs in the Coupled Model Intercomparison Project Phase 5 (CMIP5) ensemble, see Table 2. We obtain the forcing by combining the forcing data from Forster et al. (2013) and Hansen et al. (2010) such that the 18-yr moving averages of the two are equal. We use the instrumental HadCRUT dataset (Morice et al., 2012), which combines the land temperatures of the CRU dataset (Jones et al., 2012) with the sea surface temperatures of HadSST3 (Kennedy et al., 2011).

To assess the accuracy of the TCR estimations from Eq. (12) we compare the estimates from each of the 19 ESMs with the TCR obtained from the ESMs directly (Forster et al., 2013). Inference is obtained by producing 100 000 Monte Carlo simulations of the TCR. Summary statistics for our model are shown in Tables 3–4, which include the marginal posterior means and 95 % credible intervals for the TCR and the model parameters used to compute it.

To assess the approach using the scale-invariant versus the exponential response functions, we compare the posterior mean estimates with the values obtained directly from the ESMs. Specifically, we calculate the average absolute value of the bias, the root mean square error (rmse) and the correlation between the posterior mean estimates and the TCR-values from the ESMs, see Table 1. We observe that the method using the single exponential response model clearly performs worse than the other two approaches, having higher error and lower correlation with the TCR-estimates from the ESMs. The two-exponential response function and the scale-invariant approach perform almost similar, the latter approach showing slightly higher correlation. These two methods have approximately the same average absolute value of the bias, but the two-exponential approach slightly

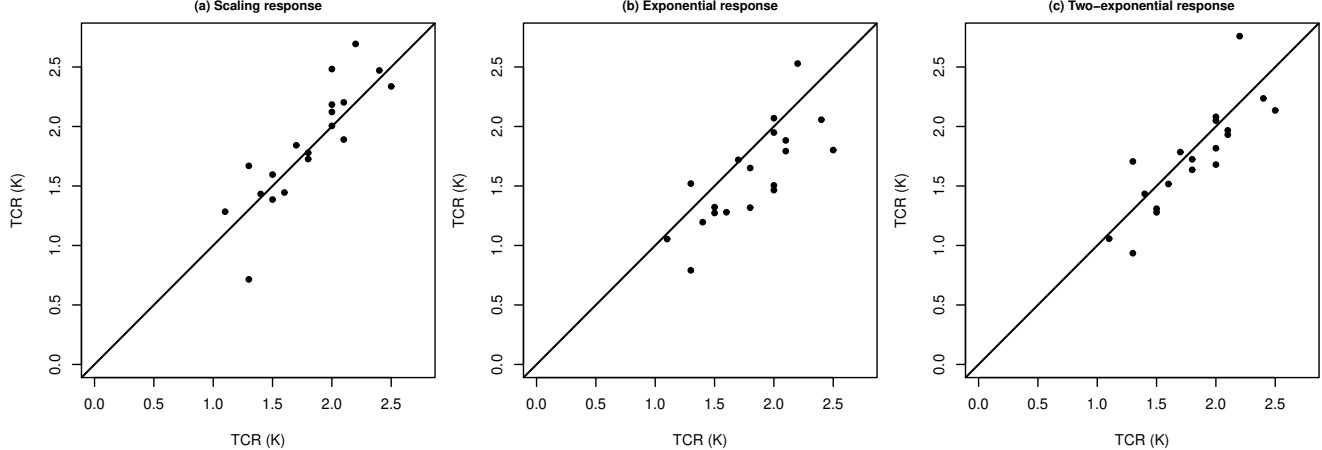

**Figure 8.** Scatter plot of the TCR obtained directly from the 19 ESMs against the corresponding marginal posterior mean estimates when using a scale-invariant response function (panel (a)), a single-exponential response function (panel (b)) and a two-exponential response function (panel (c)).

underestimates TCR while the scale-invariant approach slightly overestimates TCR. This is illustrated by the scatter plots in Fig. 8. Using `INLA.climate` we obtained, for a typical analysis, inference in around 13 seconds using a scale-invariant response and 35 seconds using the exponential response functions.

To obtain estimates for the TCR of the HadCRUT4 data set we use the 19 different forcing data associated with the ESMs
enlisted in Table: 2 as well as the Hansen radiative forcing which we will assign ID number 0. The Monte Carlo simulations are carried out separately for each forcing data set, using 100 000 samples and forcing slope coefficient $Q_{2\times\mathrm{CO}_2} = 3.8\ \mathrm{W\ m}^{-2}$ (IPCC, 2013a). This is again performed using both the scale-invariant and the exponential response functions. For the scale-invariant response, the posterior means and credibility intervals of the TCR and the parameters used to compute the TCR for each ESM are shown in Tables 5–6. The marginal posterior mean estimates and 95 % credible intervals for the TCR using
all approaches are illustrated in Fig. 9 where we observe wider credible intervals when using a single exponential response function. We obtain an estimated posterior distribution for the TCR across all models by aggregating all TCR samples obtained from each analysis, totaling two million simulations. The posterior density is obtained from the Monte Carlo samples using the `density` function in R. The resulting density function is depicted in Fig. 10, where it is compared with a histogram describing the TCRs obtained directly from the ESMs. We observe a mean of 1.43 K, 1.35 K and 1.61 K, and standard deviation of 0.19
295   K, 0.40 K and 0.40 K when using a scale-invariant, a single-exponential and a two-exponential response function, respectively. The given estimates mainly fall in the lower half of the range of TCRs and fail to capture the mode of the histogram. However, since the histogram is generated from only 19 different values of TCR its form is influenced by the size of the bins. Using bins of width 0.5K (resulting in 3 total bins) would describe a more uni-modal distribution with mode in the 1.5-2.0 interval. The posterior distribution obtained from using either a scale-invariant or the exponential response functions are still on the lower
side of this.

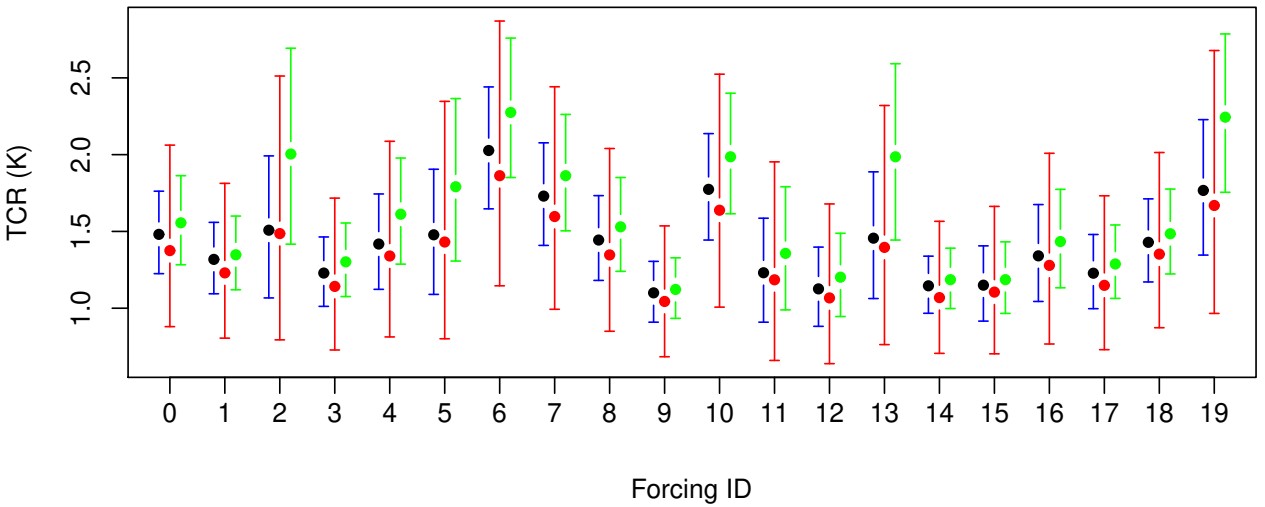

**Figure 9.** The posterior mean estimates and 95 % credible intervals of the historical TCR for all forcing data sets using a scale-invariant (blue), a single-exponential (red) and a two-exponential (green) response function. ID number 0 denotes the Hansen forcing.

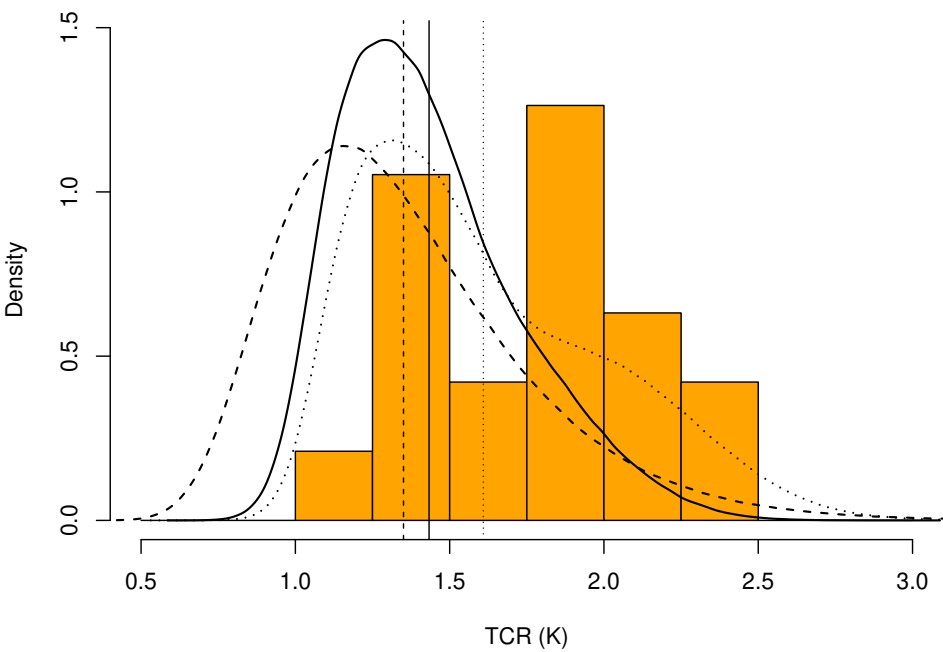

**Figure 10.** Histogram of the TCR obtained from the different ESMs with the posterior density function estimated from the Monte Carlo simulations using both a scale-invariant (solid), a single-exponential (dashed) and a two-exponential (dotted) response function. The vertical lines describe the means of the three approaches.

## 4 Conclusions

This paper presents a Bayesian formulation to analyse a linear temperature response model to radiative forcing, incorporating long-range temporal dependence using a scale-invariant response function. Computational efficiency is achieved by incorporating the model within the `R-INLA` framework and adopting the approximation introduced in Sørbye et al. (2019). The benefits of this methodology are three-fold. First, the model is both accessible and adaptable to more advanced models that require more trends and effects. Second, the approximations ensure low costs in both computational complexity and memory, even for long time series. Third, the method yields full Bayesian inference, giving a full description of the behavior of the time variables and model parameters.

In addition to providing parameter estimates, the model has been used to produce temperature predictions as responses to the four RCP forcing trajectories used to describe future radiative forcing. For comparison, we have also included prediction results using the one-box and two-box energy balance models giving single exponential or a mixture of two exponential response functions, respectively. We observe that the exponentialresponse models underestimate the predicted temperature compared to the projections made by the IPCC. On the other hand, the scale-invariant response models tend to overestimate future temperatures but are overall more accurate than using the exponential response functions.

We further demonstrate that the model can be used to estimate the transient climate response in instrumental data. Our best estimate is that the TCR is 1.43 K with a standard deviation of 0.29 K. This estimate falls right in the middle of the range put forward in the IPCC report $(0.8 - 2.5 \text{ K})$ and the accuracy is consistent with the TCR obtained directly from the ESMs. The presented model has also been seen to give coherent estimates for the equilibrium climate sensitivity, compared with running ESMs (Rypdal et al., 2018a).

Accurate linear response models for global temperature are essential alternatives to ESMs in studies where one needs to explore a large number of emission scenarios, and the modeling framework presented here can easily be included in integrated assessment models. Moreover, since the models are invertible, they can efficiently compute forcing scenarios corresponding to given future scenarios for global temperatures. Hence, in combination with linear models for the $CO_2$ response to emissions, they can be used to obtain observation-based estimates of the remaining carbon budget in scenarios where we reach the goals of the Paris agreement.

In combination with dedicated ESM experiments, the methods presented in this paper can also be used to estimate global and regional climate sensitivity as a function of background state and time scale. One can use such estimates to study the effect of non-linear responses across time scales and to obtain insight into how sensitivities and fluctuations change in the vicinity of climate tipping points.

*Code and data availability.* The code and data sets used for this paper is available through the `R`-package, `INLA.climate`, which can be downloaded from: `github.com/eirikmn/INLA.climate`.

*Author contributions.* All authors conceived and designed the study; H.-B.F. collected data and constructed the modified forcing data. E.M.-N. implemented the model, performed all of the analyses and developed the R-package `INLA.climate`. H.R provided technical support related to `R-INLA`. E.M.-N., S.H.S and M.R. discussed the results and wrote the paper.

*Competing interests.* The authors declare that they have no conflict of interest

*Acknowledgements.* The authors thank K. Rypdal for useful discussions. This project has received funding from the European Union's Horizon 2020 research and innovation programme under grant agreement No. 820970.

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

| Response function | \|bias\| | rmse | correlation |
|---|---|---|---|
| Single-exponential | 0.28 | 0.34 | 0.78 |
| Two-exponential | 0.19 | 0.24 | 0.84 |
| Scale-invariant | 0.19 | 0.25 | 0.86 |

**Table 1.** The average absolute value of the bias, the root mean square error and the correlation obtained when comparing the marginal posterior mean estimates of the TCR with the TCR obtained directly from the ESMs.

| ID | Earth System Model | Reference(s) | Time interval | TCR (K) | $Q_{2\times CO_2}$ (W m$^{-2}$) |
|----|--------------------|--------------|---------------|---------|-------------------------------|
| 1 | GISS-E2-R | (Hansen et al., 2010; Schmidt et al., 2014) | [1850,2015] | 1.5 | 3.8 |
| 2 | HadGEM2-ES | (Collins et al., 2011; Martin et al., 2011) | [1860,2015] | 2.5 | 2.9 |
| 3 | IPSL-CM5A-LR | (Dufresne et al., 2013) | [1850,2015] | 2.0 | 3.1 |
| 4 | NorEMS1-M | (Bentsen et al., 2013; Iversen et al., 2013) | [1850,2015] | 1.4 | 3.1 |
| 5 | ACCESS1.0 | (Bi et al., 2012) | [1850,2015] | 2.0 | 3.0 |
| 6 | MIROC-ESM | (Watanabe et al., 2011) | [1850,2015] | 2.2 | 4.3 |
| 7 | MIROC5 | (Watanabe et al., 2010) | [1850,2015] | 1.5 | 4.1 |
| 8 | CanESM2 | (Chylek et al., 2011) | [1850,2015] | 2.4 | 3.8 |
| 9 | CCSM4 | (Gent et al., 2011) | [1850,2015] | 1.8 | 3.6 |
| 10 | CNRMCM5 | (Voldoire et al., 2013) | [1850,2015] | 2.1 | 3.7 |
| 11 | GFDL-CM3 | (Donner et al., 2011) | [1860,2015] | 2.0 | 3.0 |
| 12 | GFDL-ESM2G | (Dunne et al., 2012, 2013) | [1861,2015] | 1.1 | 3.1 |
| 13 | CSIRO-MK3-6-0 | (Rotstayn et al., 2012; Jeffrey et al., 2013) | [1850,2015] | 1.8 | 2.6 |
| 14 | BCC_CSM 1.1 | (Wu et al., 2014) | [1850,2015] | 1.7 | 3.2 |
| 15 | BCC_CSM 1.1(m) | (Wu et al., 2014) | [1850,2015] | 2.1 | 3.6 |
| 16 | GFDL-ESM2M | (Dunne et al., 2012, 2013) | [1860,2015] | 1.3 | 3.4 |
| 17 | INM-CM4 | (Volodin et al., 2010) | [1850,2015] | 1.3 | 3.0 |
| 18 | MPI-ESM-LR | (Giorgetta et al., 2013) | [1850,2015] | 2.0 | 4.1 |
| 19 | MRI-CGCM3 | (Yukimoto et al., 2012) | [1850,2015] | 1.6 | 3.2 |

**Table 2.** The earth system models used in this paper. The table includes ID-number, references, time interval, TCR obtained directly from the ESM, and slope coefficient for the forcing corresponding to a $CO_2$-doubling.

| ID | $H$ | $\sigma_\epsilon$ (K) | $\sigma_f$ (K) | $F_0$ (W $m^{-2}$) | $T\hat{C}R$ (K) |
|---|---|---|---|---|---|
| 1 | 0.615 | 0.0546 | 0.0614 | -0.0793 | 1.39 |
|   | (0.56,0.686) | (0.0486,0.0608) | (0.0573,0.0658) | (-0.124,-0.0339) | (1.31,1.47) |
| 2 | 0.978 | 0.349 | 0.078 | 0.136 | 2.34 |
|   | (0.951,0.995) | (0.209,0.628) | (0.0602,0.0969) | (-0.0604,0.335) | (1.80,2.92) |
| 3 | 0.895 | 0.156 | 0.0775 | -0.0312 | 2.12 |
|   | (0.826,0.954) | (0.12,0.209) | (0.065,0.0899) | (-0.193,0.128) | (1.82,2.44) |
| 4 | 0.838 | 0.122 | 0.0577 | 0.081 | 1.43 |
|   | (0.758,0.912) | (0.1,0.15) | (0.0457,0.0707) | (-0.0783,0.243) | (1.16,1.72) |
| 5 | 0.898 | 0.128 | 0.0753 | 0.103 | 2.00 |
|   | (0.834,0.954) | (0.0997,0.171) | (0.0625,0.0884) | (-0.0153,0.223) | (1.68,2.35) |
| 6 | 0.857 | 0.12 | 0.0754 | 0.229 | 2.69 |
|   | (0.785,0.924) | (0.0979,0.149) | (0.0635,0.0877) | (0.0972,0.362) | (2.30,3.11) |
| 7 | 0.899 | 0.213 | 0.0443 | 0.0909 | 1.60 |
|   | (0.82,0.96) | (0.158,0.294) | (0.0298,0.0609) | (-0.252,0.446) | (1.09,2.18) |
| 8 | 0.795 | 0.155 | 0.0866 | -0.00328 | 2.47 |
|   | (0.709,0.88) | (0.131,0.184) | (0.0728,0.101) | (-0.131,0.126) | (2.13,2.83) |
| 9 | 0.743 | 0.122 | 0.0709 | -0.00588 | 1.78 |
|   | (0.655,0.834) | (0.106,0.141) | (0.0609,0.0817) | (-0.125,0.114) | (1.60,1.97) |

**Table 3.** This table contains marginal posterior means and 95 % credible intervals for the model parameters and the transient climate response, obtained from fitting the scale-invariant response model to temperature data from the first 9 ESMs.

| ID | $H$ | $\sigma_\epsilon$ (K) | $\sigma_f$ (K) | $F_0$ (W m$^{-2}$) | $T\hat{C}R$ (K) |
|---|---|---|---|---|---|
| 10 | 0.791 | 0.122 | 0.0794 | 0.12 | 2.20 |
|  | (0.72,0.865) | (0.105,0.142) | (0.0678,0.092) | (0.00457,0.236) | (1.86,2.58) |
| 11 | 0.812 | 0.136 | 0.0942 | 0.0834 | 2.18 |
|  | (0.731,0.891) | (0.115,0.164) | (0.0811,0.108) | (-0.0112,0.178) | (1.87,2.53) |
| 12 | 0.852 | 0.148 | 0.0503 | 0.249 | 1.28 |
|  | (0.77,0.927) | (0.119,0.187) | (0.0395,0.0618) | (0.0134,0.49) | (1.00,1.59) |
| 13 | 0.901 | 0.16 | 0.0734 | 0.178 | 1.73 |
|  | (0.85,0.951) | (0.132,0.204) | (0.0602,0.0854) | (0.0385,0.302) | (1.33,2.14) |
| 14 | 0.737 | 0.0862 | 0.0831 | 0.0742 | 1.84 |
|  | (0.675,0.805) | (0.0762,0.0977) | (0.0751,0.0914) | (-0.00176,0.151) | (1.71,1.98) |
| 15 | 0.755 | 0.101 | 0.0735 | 0.451 | 1.89 |
|  | (0.692,0.816) | (0.0882,0.115) | (0.0651,0.0825) | (0.331,0.575) | (1.72,2.06) |
| 16 | 0.738 | 0.154 | 0.071 | 0.0438 | 1.67 |
|  | (0.654,0.829) | (0.134,0.177) | (0.0577,0.0857) | (-0.111,0.202) | (1.38,1.98) |
| 17 | 0.918 | 0.124 | 0.0261 | 0.315 | 0.71 |
|  | (0.856,0.968) | (0.0915,0.174) | (0.0175,0.0352) | (-0.0975,0.771) | (0.51,0.95) |
| 18 | 0.903 | 0.166 | 0.0677 | -0.128 | 2.48 |
|  | (0.833,0.961) | (0.125,0.227) | (0.0569,0.0784) | (-0.308,0.0503) | (2.09,2.89) |
| 19 | 0.788 | 0.0977 | 0.0609 | 0.0214 | 1.45 |
|  | (0.708,0.868) | (0.0836,0.115) | (0.0491,0.0739) | (-0.106,0.15) | (1.18,1.74) |

**Table 4.** This table contains marginal posterior means and 95 % credible intervals for the model parameters and the transient climate response, obtained from fitting the scale-invariant response model to temperature data from the last 10 ESMs.

|   | $H$ | $\sigma_\epsilon$ (K) | $\sigma_f$ (W m$^{-2}$) | $F_0$ (K) | $T\hat{C}R$ (K) |
|---|---|---|---|---|---|
| 0 | 0.817 | 0.118 | 0.05 | 0.32 | 1.48 |
|   | (0.74,0.891) | (0.0995,0.142) | (0.0402,0.0606) | (0.132,0.51) | (1.22,1.76) |
| 1 | 0.807 | 0.116 | 0.0455 | -0.0159 | 1.32 |
|   | (0.726,0.883) | (0.0976,0.138) | (0.0367,0.055) | (-0.217,0.189) | (1.09,1.56) |
| 2 | 0.932 | 0.194 | 0.0554 | 0.413 | 1.51 |
|   | (0.877,0.975) | (0.141,0.283) | (0.039,0.0733) | (0.15,0.692) | (1.07,1.99) |
| 3 | 0.821 | 0.119 | 0.0509 | 0.0409 | 1.23 |
|   | (0.744,0.895) | (0.1,0.143) | (0.0409,0.0617) | (-0.145,0.23) | (1.01,1.46) |
| 4 | 0.863 | 0.135 | 0.0548 | 0.184 | 1.42 |
|   | (0.791,0.927) | (0.109,0.169) | (0.0424,0.0684) | (-0.016,0.389) | (1.12,1.74) |
| 5 | 0.917 | 0.173 | 0.0538 | 0.38 | 1.48 |
|   | (0.858,0.965) | (0.13,0.239) | (0.0394,0.0695) | (0.136,0.637) | (1.09,1.90) |
| 6 | 0.841 | 0.125 | 0.0584 | 0.228 | 2.03 |
|   | (0.769,0.91) | (0.104,0.153) | (0.0463,0.0714) | (0.057,0.403) | (1.65,2.44) |
| 7 | 0.835 | 0.124 | 0.0528 | 0.191 | 1.73 |
|   | (0.762,0.906) | (0.103,0.15) | (0.0419,0.0645) | (0.00416,0.381) | (1.41,2.08) |
| 8 | 0.835 | 0.124 | 0.0478 | 0.139 | 1.44 |
|   | (0.762,0.907) | (0.104,0.15) | (0.038,0.0583) | (-0.0649,0.347) | (1.18,1.73) |
| 9 | 0.81 | 0.118 | 0.0399 | 0.0653 | 1.10 |
|   | (0.73,0.888) | (0.0991,0.141) | (0.032,0.0483) | (-0.165,0.302) | (0.91,1.31) |

**Table 5.** This table contains marginal posterior means and 95 % credible intervals for the model parameters and the transient climate response, obtained from fitting the scale-invariant response model to the HadCRUT dataset using forcing data from Hansen et al. (2010) (denoted by ID 0) and from the first 9 ESMs.

| | $H$ | $\sigma_\epsilon$ (K) | $\sigma_f$ (W m$^{-2}$) | $F_0$ (K) | $T\hat{C}R$ (K) |
|---|---|---|---|---|---|
| 10 | 0.838 | 0.125 | 0.0597 | 0.201 | 1.77 |
| | (0.764,0.908) | (0.103,0.152) | (0.0472,0.0732) | (0.0335,0.372) | (1.44,2.14) |
| 11 | 0.906 | 0.166 | 0.0456 | 0.579 | 1.23 |
| | (0.844,0.959) | (0.127,0.224) | (0.0332,0.059) | (0.289,0.885) | (0.909,1.59) |
| 12 | 0.865 | 0.138 | 0.0434 | 0.356 | 1.13 |
| | (0.793,0.93) | (0.112,0.174) | (0.0332,0.0545) | (0.0901,0.631) | (0.882,1.4) |
| 13 | 0.927 | 0.184 | 0.0601 | 0.309 | 1.46 |
| | (0.871,0.972) | (0.136,0.261) | (0.0437,0.078) | (0.0834,0.546) | (1.06,1.89) |
| 14 | 0.788 | 0.111 | 0.0483 | 0.0442 | 1.15 |
| | (0.707,0.869) | (0.0951,0.131) | (0.0394,0.0578) | (-0.131,0.222) | (0.97,1.34) |
| 15 | 0.851 | 0.131 | 0.0391 | 0.183 | 1.15 |
| | (0.777,0.92) | (0.107,0.163) | (0.0304,0.0484) | (-0.0866,0.463) | (0.92,1.41) |
| 16 | 0.865 | 0.139 | 0.0474 | 0.0836 | 1.34 |
| | (0.792,0.932) | (0.112,0.177) | (0.0358,0.06) | (-0.166,0.339) | (1.04,1.67) |
| 17 | 0.833 | 0.124 | 0.0517 | -0.167 | 1.23 |
| | (0.756,0.905) | (0.102,0.15) | (0.0411,0.0633) | (-0.361,0.0319) | (1.00,1.48) |
| 18 | 0.826 | 0.122 | 0.0444 | 0.0684 | 1.43 |
| | (0.749,0.901) | (0.102,0.147) | (0.0353,0.0542) | (-0.149,0.29) | (1.17,1.71) |
| 19 | 0.895 | 0.153 | 0.0631 | 0.0303 | 1.77 |
| | (0.831,0.952) | (0.12,0.202) | (0.0474,0.08) | (-0.164,0.231) | (1.35,2.23) |

**Table 6.** This table contains marginal posterior means and 95 % credible intervals for the model parameters and the transient climate response, obtained from fitting the scale-invariant response model to the HadCRUT dataset using forcing data from the last 10 ESMs.