# Peer review of "Statistical estimation of global surface temperature response to forcing under the assumption of temporal scaling"

_Earth System Dynamics, 2019_

## Referee Comment (RC1) · Anonymous Referee #1 · 19 Dec 2019

General comments

The paper describes a new stochastic model the response of global mean surface temperature to radiative forcing which assumes a scale-invariant response function instead of a combination of exponential functions. The model is fitted to temperature and radiative forcing data from the GISS-E2-R, and further used to forecast temperature for different CO2 forcing scenarios and to estimate the transient climate response (TCR). The results of the scale-invariant model are compared with the ones resulting from an exponential response function. The paper seems in my opinion scientifically sound and the results are interesting from the applications point of view. It's not clear to me,

however, whether the comparison is performed with a 1-box energy model or with a combination of m exponential functions (and in that case what is the size of m).

Specific comments

- Please clarify the meaning of "emergent" (emergent symmetry, pg 3, line 58; emergent property, pg 3, line76).

- Maybe provide a reference for the Goddard Institute for Space Studies (GISS) E2 model (page 4, line 85).

- Fig 8: maybe explain why both the scale-invariant and exponential curves are unable to capture the histogram mode.

- Fig. 8, caption: maybe remove the "dashed" in the description of the vertical lines (for the mean of the exponential response function).

---

## Author Comment (AC1) · 30 Dec 2019

We appreciate the comments from the reviewer, and feel they make several valid points. Thus, in the revision we will change the text to accommodate the following:

1. The comparison is performed with a single exponential response function corresponding to a 1-box energy model, against a scale-invariant response function. This will be made more clear in the revised manuscript.

2. Temporal scale invariance in global temperature fluctuations is an empirical observation and not deduced from the fundamental equations describing the system. Nevertheless, climate models ubiquitously exhibit scale invariance, suggesting that this is a complex-systems phenomenon. It emerges from the interaction between many dynamical components.

Since this discussion is slightly beside the main objective of the current paper, we suggest removing the word "emergent" from the sentence.

3. A reference for the GISS-E2 model will be added.

4. Since the histogram is generated from only 19 different values of TCR (corresponding to the ESMs examined in this paper) its form is highly influenced by the size of the bins. Using bins of width 0.5K (resulting in 3 total bins) would describe a more unimodal distribution with mode in the 1.5-2.0 interval. The posterior distribution obtained from using either a scale-invariant or an exponential response function are still on the lower side of this however. We will comment on this in the revised manuscript.

5. The "dashed" term is used to distinguish between the density and mean of the posterior using an exponential response function from a scale-invariant response function. If this clarification is not needed we can remove it.

---

## Referee Comment (RC2) · Anonymous Referee #2 · 4 Jan 2020

This is an interesting and useful piece of work which is well worth publishing. While physically-based models are attractive in terms of interpretation of parameters and underlying mechanisms, if a model that is not so directly intepretable can be shown to perform as well or better, then as well as being functionally useful that's an additional challenge and avenue for improving our understanding. The parameter estimation method also looks like a powerful approach and demonstrating its use may be of benefit for many readers.

I am however a little bit underwhelmed by the comparison to the simple exponential model. At least one of the authors has already shown that this is far from adequate for

modelling the transient behaviour over the 20th century! Thus, it's really too easy a target to beat. While the point about additional parameters is well made, adding a second deep ocean layer is surely not too demanding and such a model has markedly better performance. If the authors don't want to include additional uncertain parameters, those relating to the second layer could even be fixed at plausible values.

There is also a bit of a gap in the analysis that really needs to be filled. Unless I have missed something, there is no direct analysis of how well the model performs at predicting GCM behaviour. The analyses in Fig 4 and 5 fit the models to observed temperatures and compare projections to the models, observing that the simple exponential model tends to underpredict compared to the GCM-based AR5 projections. It would be more of a test to fit the models to each of the GCM hindcasts and see how well they manage to predict the respective GCM futures under a given scenario. While the scale-invariant model appears to match the projections better, the reason that the exponential model underestimates the projections may partly be that the GCMs overestimate the (transient) response to forcing. Though as mentioned above the relatively performance of the exponential model is not particularly noteworthy anyway.

I'd also like to see more discussion of the striking difference between the two panels in Fig 2. What is the underlying explanation for the different uncertainties, and are the estimated uncertainties in 2(a) reliable? Conversely, the uncertainties in Figs 4 and 5 appear the other way round, with the scaling response leading to much larger spread. Error bars on the TCR estimates on Fig 6 would also be helpful.

---

## Author Comment (AC2) · 31 Jan 2020

We thank the reviewer for the comments. We agree that the 1-box model is not an adequate benchmark for the model we use. Our idea was to use it more for illustration since it is a well-known simple climate model. In our revised manuscript, we will make this clearer. Moreover, we will incorporate a 2-box model to the analysis, as suggested by the reviewer.

The second main point the reviewer makes is that we should compare the model's temperature projections to ESM temperature projections (under RCP scenarios) after we have tuned the statistical model historical runs of the corresponding EMSs. We

agree that this is a reasonable way of testing the model, and we will incorporate the analyses in the revised manuscript. The results show that the predictions made using the statistical model slightly overestimate the temperature increase in the ESMs. This overestimation is not a statistical bias. Instead, it shows that scale invariance is too crude an approximation for several ESMs. However, not all climate models have scaling properties consistent with temperature reconstructions, and hence one should be careful in how one interprets the apparent overestimation.

Response to the last paragraph in the review:

The uncertainties illustrated in figure 2 describe only the uncertainty of the estimated forced response. The remaining noise term also affects the temperature.

The explanation for the difference in uncertainties in the panels of figure 2 is simply that noise-free response to the known forcing gives a much better fit for the scaling model compared to the model with an exponential response model. It is an illustration of the well-know inadequacy of the 1-box model.

The uncertainty bars in figures 4 and 5 describe the total uncertainty of the model, not just that of the forced temperature response, which we show in figure 2. We will make this clearer in the revised manuscript.

We will also add error bars for figure 6.

---

## Author Response (AR1)

**Pointwise reply to reviewers and list of changes we've made to the manuscript**

Reviewer 1:

1. We have made the text associated with figure 2 (old manuscript) more clear in that w are comparing the forced response from fitting a single exponential response function to a scale-invariant response function.
2. We have removed the use of "emergent".
3. We have added a reference for the GISS-E2 model.
4. We have added commented on the histogram of figure 8 (old manuscript).
5. We have left "dashed" term in the caption of figure 8 (old manuscript) as we feel it helps distinguish between the different approaches.

Reviewer 2:

1. We have made the text more clear in that the one-box model is not used as a benchmark, but as illustration.
2. We have added a two-box model to the analysis, see figure 2, 6, 8, 9 and 10 (new manuscript).
3. We have performed temperature predictions using ESM temperature projections under RCP scenarios and have compared these predictions with the ESM runs in figure 7 (new manuscript), which show the deviation between them.
4. We have changed the text to clarify that the uncertainties of figure 2 (old manuscript) only describes the uncertainty of the forced response, and not the unforced response. On the other hand, figures 4 and 5 (old manuscript) show the total uncertainty of the model (both responses), the text has been updated to clarify this.
5. We mentioned adding error bars for figure 6 (old manuscript). However, the line in the figure is not produced by the data and is only meant to help illustrate over- and underestimation. We have therefore left it in unchanged. We have also flipped the axes of the figures to make points representing overestimations lie above the line.

[revised manuscript text omitted]